# Does the Physiological Response of a Triathlete Change in the Use or Absence of Drafting?

**DOI:** 10.3390/ijerph19159366

**Published:** 2022-07-30

**Authors:** David Mancha-Triguero, Pablo Pérez-Murillo, Sergio J. Ibáñez, Antonio Antúnez

**Affiliations:** 1Department of Physical Education and Sport, Cardenal Spínola CEU Andalucía University, 41930 Sevilla, Spain; dmancha@ceuandalucia.es; 2Group for Optimisation of Training and Sport Performance (GOERD), Faculty of Sport Science, University of Extremadura, 10003 Cáceres, Spain; pperezmu@alumnos.unex.es (P.P.-M.); antunez@unex.es (A.A.)

**Keywords:** sprint triathlon, drafting, physiological responses, performance, tactics

## Abstract

Background: Currently, tactics play an important and decisive role in sprint distance triathlons. One of the most decisive tactical elements is drafting in the cycling sector, depending on whether or not it is allowed by the test regulations. The objective was to analyze the physiological responses in running, in relation to drafting in the cycling sector, according to level and sex. Methods: To do this, a total of *n* = 44 subjects were divided into two levels (elite: they got a podium in the national championship, 15.68 ± 0.82 years; amateurs: they compete at the regional level, 15.68 ± 1.62 and 37.9 ± 1.74 years), undergoing two training sessions of four cycling-running multitransitions with variability in the permissibility of drafting were analyzed. A descriptive analysis of the variables was carried out, together with an inferential analysis to know the relationships and associations between the dependent and independent variables. Results: The results showed significant differences in the parameters, as related to running technique, heart rate, speed, and displacement (both between levels and sex). Conclusions: This study concludes that drafting in the cycling sector generates decisive physiological responses for the running sector.

## 1. Introduction

Triathlon is an Olympic sport that involves the performance of three sports disciplines (swimming, cycling, and running), which are carried out in order and without interruption, thus offering a wide variety of disciplines and distances [1]. According to current literature, the sprint distance is the one that enjoys the greatest popularity among popular triathletes, or age groups, because it requires less physical demand and can be within the reach of the majority of the population, without needing volumes of high training, such as the middle- and long-distance, which are difficult to combine with work and social life for amateur athletes [2].

As a sport, a triathlon encompasses different modalities collected under the regulations of the Spanish Triathlon Federation, also known as FETRI [3], such as duathlon, cross duathlon, winter triathlon, or quadriathlon, among others. All these official competitions are organized according to the category of the participant. The regulation has an absolute male category and another absolute female category. In turn, these categories (according to sex) are divided according to the age of the participants [2].

All these modalities that are grouped within the triathlon, have a set of common norms or rules, such as the use of drafting or the distance of the test, which is related to the age of the participants. Younger participants will only be able to participate in short-distance events, while adults can participate in both short- and long-distance [3]. To do this, depending on the test to be performed, the preparation of the athlete will be modified. In this line, training planning is of great importance, since it must be oriented and based on the training principles: (i) supercompensation; (ii) overload; (iii) specificity; (iv) individualization; (v) variation; (vi) recovery; (vii) progression; and (viii) reversibility [4]. In addition, one of the main objectives of planning is related to the quantification of the load. The quantification of the load is the method used to know the volume and intensity to which athletes are subjected, thus being vital to achieve the objectives and physiological adaptations proposed in any sports planning [5].

Good load control and correct interpretation provide the coach with a set of reliable information regarding how the training process is affecting the athlete [6,7] and if the load planned for the athlete based on their needs and characteristics is causing the necessary or planned adaptation [8]. Traditionally, individual sports have focused part of their efforts on the control and quantification of the loads that these athletes support during training or competitions, with the aim of improving and planning training [9]. This trend has spread to other sports, currently being a booming issue in recent years [10]. However, in a large part of individual sports, this process has hardly changed, and technology can help both in the evaluation and quantification of daily work to optimize performance [11]. In this line, training groups are often seen, in which athletes with different objectives, moments of the season, age, or even competitive level are mixed; training in situations far from those that can be found in competition. In addition, in most competitions there is little knowledge of the competition that does not allow either coaches or athletes to predict the final result [12]. To do this, the tactic or approach of the competition is used. In this part, it is necessary to attend mainly to the regulations and, from there, look for a situation that most favors the athlete. In recent years, one of the trends in triathlons has been the use or prohibition of drafting, defining this fact as the situation of an athlete who during the competition is close to another cyclist who is ahead and takes advantage of his situation to reduce his physical–physiological demands to gain an aerodynamic advantage [13] (benefit from a possible decrease in resistance), and this variant may alter the final result of the competition. For this, the reproducibility of this situation in the training sessions in the organization of the competition is very important. This variant can alter the final result in the competition, and the reproducibility of this situation is very important in training, in order to organize the competition.

We reviewed the literature, and there are documents that analyze the different modalities of triathlon (in a generic way) or some specific sector [6,7,8,9,10]. However, there are no documents that analyze the influence of the cycling sector on the running sector and how drafting affects the triathlete’s fatigue and technique. For all these reasons, the objectives of this research were: (i) to characterize the physical demands that triathletes endure in multi-transition training; (ii) analyze the differences in the demands they support according to the competitive level, sex, and type of circuit (with drafting/without drafting), according to the origin of the selected variables (internal load and external load variables).

## 2. Materials and Methods

### 2.1. Design

Following the classification model established by this research, we can frame it under a manipulative strategy, within studies of an empirical nature of a quasi-experimental nature, where we seek to examine the differences between groups of athletes, being a non-equivalent group design [14].

### 2.2. Sample

The sample of participants selected for this research corresponds to the non-random type, characterized by the influence of the researcher who selects the sample, and the choice made may be due to different factors. Within this group, it belongs to the intentional or opinion types, where the sample is chosen based on the study to be carried out and where representative samples are sought. A total of 44 triathletes (*n* = 44) included between the young category and the senior age group, both male and female, were analyzed (Table 1).

All the selected triathletes train and compete regularly. On the one hand, there is the performance group made up of high-performance triathletes (they are the best triathletes in their categories—in young triathletes, they obtained a podium in the national championship; in adults, triathletes placed 15th or better in the national championship). On the other hand, the amateur/amateur group corresponds to triathletes from different clubs that compete in the regional triathlon league.

### 2.3. Materials and Instruments

Anthropometric characteristics were recorded using a portable rod stadiometer (SECA, Hamburg, Germany) and bioimpedance balance to record body composition using 8 model MC-780MA electrodes (TANITA, Tokyo, Japan). To record the variables related to the load borne by the athletes, each athlete was equipped with a WIMU PROTM inertial device (RealTrack Systems, Almería, Spain), which was attached to the athlete’s body by means of an anatomically adjustable harness for practice, as well as a fitness tracker and GARMIN heart rate strap (Garmin, Olathe, KS, USA). The inertial device was used in the analysis and recording of the kinematic variables of distance and time, of accelerometry and neuromuscular. The heart rate band was for variables related to objective internal load (heart rate). To know the beginning and end of each athlete’s test, an ANT+ transmitter was used that incorporated a button to mark the start and end points on the timeline of the devices [15].

### 2.4. Variables

The load variables are those that allow the coach to have a quantification of the demands that training causes in the athlete. Within the variables, one can differentiate between independent and dependent variables. In this research, the independent variables were sex (men and women), competitive level (amateur/amateur and elite), and circuit type (with and without drafting). The rest of the variables, defined below, are dependent:
*Variables related to the cycling sector:*(i)Average speed cycle sector (avg speed cycle): the average speed at which the athlete moves during his cycling phase, measured in km/h.*Variables related to the running sector:*(ii)Time: time used by the athlete to cover the estimated distance of the running phase (400 m), measured in seconds [11,16,17].(iii)Explosive distance (Dist. Expl.): number of meters traveled by the athlete at a speed greater than 18 km/h over a distance of 400 m, measured in meters.(iv)Distance traveled at different intensities: number of meters that the athlete travels at different speeds. The ranges are 0–15 km/h; 15, 1–20 km/h; 20, 1–25 km/h; more than 25 km/h, measured in meters.(v)Maximum heart rate (HRMax): maximum number of beats made by the heart in one minute, measured in beats per minute (bpm).(vi)Average heart rate (HRAvg): average number of beats made by the heart in one minute, measured in beats per minute (bpm).(vii)% Maximum heart rate (% HRMax): percentage at which the heart works, taking the maximum heart rate into account. It is a variable of intensity.(viii)Average speed (Speed avg): average speed at which the athlete has moved in a period during the running phase.(ix)PlayerLoad (PL): a vectorial magnitude derived from triaxial accelerometry data that quantifies motion at high resolution. Accelerations and decelerations are used to construct a cumulative measure of the rate of change in acceleration. Cumulative measure (PL) and intensity measure (PL·min^−1^) were used, thus being able to indicate the rate of stress to which the player submits his body during a given period of time. As a unit of load, it has a moderate-high degree of reliability and validity [17,18].(x)Number of steps (nStep): count of steps/strides made during the running phase. A smaller number of steps informs that the stride is wide. Measured in number of times.(xi)Step time (tStep): total time the athlete spends to perform a full step cycle. This pitch cycle is made up of the ground contact time and the flight phase time spent moving, measured in milliseconds.(xii)Contact time (tContact): the time that the athlete is in contact with the ground during the step and in which he tries, in the shortest possible time, to print the greatest possible force in the movement, measured in milliseconds.(xiii)Step flight time (tFly): the time that the athlete is in the flight phase after the contact time. Depending on the contact time variable, the flight phase can be longer or shorter, measured in milliseconds.*Variables related to neuromuscular fatigue of the test:*(xiv)CMJ Jump height (CMJ): maximum height reached during the CMJ jump, measured in centimeters [19].


The variables HRMax, HRAvg, % HRMax PL, and CMJ were used in other similar studies for the quantification of internal and external load [20,21].

### 2.5. Statistical Analysis

A descriptive analysis was performed for the analyzed variables (*mean and typical deviation*). Next, the criteria assumption tests were carried out through the normality assumption (*Shapiro–Wilk*), homoscedasticity assumption (*Levene test*), and randomization assumption (*Runs test*), finding a distribution of the data as parametric [22]. In addition, an inferential analysis was performed to determine the differences based on the independent variable. For this, the selected test was *t test for independent samples* [23]. The value of significance used in the investigation was *p* < .05, according to the specification of Field [22]. The software used was the statistical package for the social science (SPSS Statistics, version 24, IBM Corporation, Armonk, NY, USA).

### 2.6. Procedure

Firstly, the territorial triathlon federation, where the investigation was going to be carried out, was contacted. Once the proposal was accepted, the technicians involved were informed of the research design (some of their triathletes could be part of the final sample). In addition, for minor triathletes, the parents were informed, who, through an informed consent, authorized the participation of their children in the research. Next, a contact was made with the athletes, so that they could experience the investigation procedure and materials that they were going to wear.

The study consisted of performing two multi-transition training sessions with a separation of 2 weeks (the same training strategies, nutrition, recovery, and accumulated weekly training load were maintained, an attempt was made to reproduce them in both evaluations). Each multi-transition training will be divided into four cycling and running transitions, with a total recovery of 10 min between each transition, in order to analyze the parameters with the least possible wear. Each cycling circuit will have a distance of 2000 m, and each running circuit will have a distance of 400 m. Both circuits will take place on asphalt, with the aim of reproducing competition conditions. In test 1, drafting was prohibited. Each triathlete started individually, with a margin of 1 min between the next subject to start the transition.

In test 2, drafting was allowed. The outputs were achieved in groups of 10, in order to reproduce competition situations. The circuits used in both evaluations were the same, so that the circuit was not a polluting variable. In each training, the triathletes used the same sports material and nutrition and hydration system. In addition, the day chosen for the test was the same day of the week, and the load accumulated during the week in both situations was similar.

## 3. Results

The results of the investigation, according to the independent variables, are presented below.

### 3.1. Differences between Sex without Drafting

Table 2 below shows the descriptive results and significant differences between sex in the circuit without drafting, as differentiated in each of the four transitions of the elite athletes. In transition 1, significant differences are shown in the distance traveled at different speeds, as well as the flight time related to the running technique. In transition 2, significant differences are shown in the distance covered at different speeds and jump height CMJ, related to fatigue. In transitions 3 and 4, significant differences are shown in the distance traveled at different speeds. Both in transitions 2, 3, and 4, no significant differences were observed in the variables related to running technique for elite level athletes.

It also analyzes the differences between sex in the circuit without drafting at the amateur level divided into the four transitions. In transition 1, significant differences were found in the distance traveled at different speeds, explosive distance, heart rate variables, and PlayerLoad. In transition 2, significant differences were found in the distance traveled at different speeds, heart rate variables, and average running speed. In transition 3, significant differences were found in the variables of heart rate, PlayerLoad, and average speed of the running phase. In transition 4 significant differences were found in the distance traveled at different speeds. In addition, in the 4 transitions carried out, significant differences between sex were observed in the amateur athletes in the variables related to running technique.

### 3.2. Differences between Sex with Drafting

Table 3, below, shows the descriptive results and the significant differences between sex in the circuit with drafting, as differentiated in each of the four transitions of the elite athletes. In transition 1, significant differences are shown in the explosive distance, heart rate variables, and PlayerLoad. In transition 2, significant differences are shown in the distance traveled at different speeds, average speed of the running phase, and PlayerLoad. In transitions 3 and 4, significant differences are shown in the distance traveled at different speeds, heart rate variables, average speed of the running phase, and PlayerLoad. In addition, in transitions 1, 2, and 4, significant differences were observed in the variables related to running technique for elite-level athletes.

The differences between sexes in the circuit with drafting at the amateur level divided in the four transitions were also analyzed. In transition 1, significant differences were found in the distance traveled at different speeds. In transitions 2 and 3, no significant differences were found in the variables analyzed. In transition 4, significant differences were found in the number of steps made. Finally, in this modality with drafting, differences were observed at the elite level in most transitions when fatigue was evaluated (through the CMJ jump).

### 3.3. Differences with Drafting/without Drafting between Competitive Levels

Table 4 shows the results grouped according to the competitive level and use, or absence, of drafting during the four transitions. The results belonging to elite level athletes show that there are significant differences in the parameters of distance traveled at different speeds in transitions 1, 2 and 4. In addition, significant differences are shown in the variables related to the running technique (stride) in variables 2, 3, and 4. On the contrary, in the amateur-level subjects, significant differences are shown in the parameters of distance traveled at different speeds in transition 1. Finally, significant differences are also shown in the variables related to the running technique (stride) in variables 2, 3, and 4.

## 4. Discussion

The analysis and quantification of the load in training or competition in individual sports has been a common practice for many years. However, it is currently still evaluated or quantified with the same procedure, and very few incorporate new technology or parameters. In addition, on many occasions, the athlete is evaluated without paying attention to the test or format for which it is intended to prepare. That is why the characteristics of the test must have an impact on the preparatory process.

### 4.1. Differences between Sex in the Circuit with Drafting

The significant differences of the analyzed variables, according to sex in the circuit with drafting in elite-level athletes, showed the main finding of the research that the running technique and physiological variables vary heterogeneously, according to sex, with fatigue. These differences especially appear in the variables jump CMJ (assessment of fatigue) and the flight phase of the stride. In this line, Reilly [24] analyzed the difference in running technique between women and men in athletes. This study concluded by stating that the anatomical differences between a woman and man affect the development of the running technique. One of the most prominent causes is due to the fact that women have a wider pelvis than men, which affects the “Q angle” between the pelvis and femur [25,26]. This study also suggested that women’s pelvises are more tilted than men’s, inducing a forward curvature when running. The fact of not finding significant differences in the rest of the parameters is attributed to the sports level of the athletes analyzed, since, at these levels, women train at high volumes and intensities that allow them to perform in competitions at a high aerobic and anaerobic levels [27].

However, if the significant differences in the test performed by amateur-level athletes are analyzed, the main findings are the difference in the running technique in the flight phase (measured through ‘tFly’) that affects the difficulty to reach and maintain high running paces. Differences in running technique can be attributed to the anatomical differences in women, compared to men, as discussed above [25,26]. To analyze the differences in reaching high rhythms, the existing literature shows different studies related to specific training for women [28]. Along these lines, García [29] analyzed the physiological differences between men and women that must be taken into account when planning training sessions. According to this study, women should train for anaerobic thresholds at higher intensities than men, since their fat metabolism is higher and maintained at higher intensities than men. In addition, he suggested that, if men normally enter anaerobic consumption at 80–85% of their maximum heart rate, women should do so at 85–90% of their maximum heart rate, in the same way they are training in mixed aerobics instead of anaerobic. For all these reasons, different causes have traditionally been shown that made it difficult for women to reach high running rhythms, with the planning being carried out with a generic design and without taking their physiological or hormonal considerations into account, compared to those of men [30]. The fact that no significant differences were found in the rest of the parameters is related to the fact that, although the level is amateur, they are triathletes who compete regularly, and their energy systems are developed and stimulated to perform intense training [27,31,32].

### 4.2. Differences between Male/Female Sex in Circuit without Drafting

In the test carried out on the circuit without drafting, the elite athletes analyzed responded differently to the same stimulus, depending on their sex. Although there are differences related to running technique and anaerobic threshold, the main finding is the appearance of significant differences, as related to the other parameters of running technique and displacement (meters traveled at different speeds or step contact time), as well as the high % of maximum heart rate achieved. Coinciding with the findings mentioned above, the anatomical differences in women cause a change in the running technique, with respect to men, as well as the fact that our affected technical parameters appear can be associated with the fact that the wear associated with a circuit without drafting is higher than in a circuit with drafting. In this line, Melendo [33] stated that the energy wear of going to the wheel (drafting) is equivalent to a savings of 30%, due to factors related to aerodynamics. Concurring with the statement, Anderson [34] suggested that, in addition to aerodynamics, muscle mass also interferes with energy wasting because the greater the muscle mass, the greater the displacement and rolling force. According to body composition, women in this study have a higher percentage of muscle mass than men. This fact would justify a greater energy expenditure by women in this type of circuit that could harm the technical execution of the subsequent foot race [35,36,37]. In addition, the appearance of significant differences in the variables related to heart rate is also related to fatigue, body composition, and the wear and tear produced by the absence of writing [38]. This can generate poorly planned training (generic, without differentiating by sex or according to physiological factors), as well as training without being structured according to the woman’s anaerobic threshold, which can cause greater physical exhaustion, thus justifying the findings obtained in this manuscript.

### 4.3. Differences in the Use/Absence of Drafting between Competitive Levels

Finally, the differences between elite and amateur athletes in the use or absence of drafting in physical demands have been analyzed. The main finding is the appearance of significant differences at both levels in parameters, as related to cardiac output, running technique (stride), and movement speed. To do this, Gutiérrez [39] analyzed the aerodynamic losses in cycling, based on the number of cyclists that make up a group and concluded that the fact of going in a group of five cyclists or more “on the wheel” implies an energy waste between a 20–30% less than in a circuit without drafting, where the mechanical and aerodynamic losses are greater. This fact justifies the differences in cardiac wear between one circuit and another. In addition, in relation to the variables, Calas [40] analyzed the biomechanical characteristics of different triathletes. This study concluded that there was previous wear in the swimming and cycling segments conditions running technique. This fatigue tends to make triathletes execute a pendulum running technique, instead of a circular one [41]. Caro [42] stated that the fatigue generated by swimming and cycling makes the arms swing less. In turn, the fatigue in the hamstring muscles generated in the cycling segment makes the support time longer and flight phase more less accentuated [32,43]. In line with the findings found, Coles et al. [44] determined that, in cycling situations where drafting is not allowed, oxygen consumption increases between 18–33% and lactate accumulation appears earlier than it does on circuits without drafting. This fact justifies the differences found in the high heart rate intervals that require a greater energy expenditure. Another study, Castellar [45] suggests a greater specialization of training, giving greater importance to race tactics, as well as an individualization of the race technique, depending on the competition circuit, and greater differentiation in training between men and women. The variation in the explosive distance in running is due to the muscular and physiological fatigue generated in the cycling sector [46]. According to Mon [47], the muscular fatigue generated by swimming and the cycling sector generates a loss of explosive force of 15% in international triathletes. These data are associated with the previous study [44], where energy and physiological expenditure without drafting is increased, which could justify this difference in explosive distance between the circuit with and without drafting.

This study provides a new point of view on the importance of good race tactics, in order to perform at the highest level in a triathlon. The fact that the cycling segment does not allow drafting has clear and direct consequences for the subsequent running race, regardless of the level of the triathlete. Increasingly, the ITU (International Triathlon Union) is betting on short and explosive test formats, where every second will be key to achieving good results. Having relevant data regarding how to improve and enhance running biomechanics and reduce cardiac output can be key in this type of test format. In turn, COVID-19 has generated some changes in the regulations, where many competitions were held with the individual time trial format where drafting was not allowed. This study highlights the increasingly pronounced importance of working on racing technique, according to the needs of the circuit and in an individualized way, starting from the lower categories. In addition, it provides useful information on the need for training and practical application of specific training differentiated for women and men. Training at the same intensities and structures as men has been shown to impair cardiac performance in demanding training and future competitions. This study suggests the need for greater training and specialization of coaches in the specific training of female athletes.

Inertial device technology has been used in a novel way, since there are hardly any cyclical sports studies that have used it, and it has served to provide clear information on biomechanical aspects of running on a field test and in a situation close to competition. This information facilitates the planning or detection of any anomaly, due to the fact that it has traditionally had a high cost, and not all athletes can access it. Finally, the application of this technology is suggested to carry out assessments and monitoring of cyclical sports, in order to optimize their performance in training and competitions. It also suggests working on the running technique in men differently than in women, taking the anatomical differences that they present into account. In addition, a limitation of the research is the number of the sample (*n* = 44), which is limited and has specific characteristics. It would be interesting to, in future research, expand the number of athletes analyzed (expanding in amateur and elite, both male and female).

## 5. Conclusions

The main conclusions confirm that the physiological demands supported are influenced by the sex, type of circuit, and level of the athlete. It is also concluded that drafting influences cardiac output in elite and amateur running. For this reason, it is concluded that the circuit without drafting generates a loss of explosive force in the running race, as reflected in the explosive distance and height of the jump. Additionally, drafting has a greater influence on women than men, based on steps/stride-related parameters (flight time, step time, and contact time). Drafting influences cardiac output in elite and amateur runners.

For all these reasons, taking the results obtained in the research into account, it is recommended that coaches and sports professionals, during the planning and development of training sessions, seek to prepare the athlete (regardless of sex or sports level) for the sport, physical activity, or effort that will be made during the competition—often, training groups are observed doing joint training (with the help of drafting), when, during the competition, that “extra help” is prohibited.

## Figures and Tables

**Table 1 ijerph-19-09366-t001:** Physical and biological characteristics of the selected triathletes.

Group	Years	Sample	Triathletes	Characteristics
Young elite	15.68 ± 0.82	*n* = 14	*n* = 8 males; *n* = 6 females	Males: Height: 167 cm; Weight: 53 kg; Fat: 7.6%. Females: Height: 162 cm; Weight: 47 kg; Fat: 8.1%.
Young amateur	15.68 ± 1.62	*n* = 14	*n* = 7 males; *n* = 7 females	Males: Height: 154 cm; Weight: 43 kg; Fat: 8.2%. Females: Height: 150 cm; Weight: 40 kg; Fat: 8%.
Amateur	37.9 ± 1.74	*n* = 16	*n* = 8 males; *n* = 8 females	Males: Height: 175 cm; Weight: 73 kg; Fat: 13.3%. Females: Height: 171 cm; Weight: 63 kg; Fat: 15.2%.

**Table 2 ijerph-19-09366-t002:** Descriptive and inferential results of the variables analyzed based on the circuit transition without drafting.

			Elite						Amateur					
			Men		Women				Men		Women			
			*Mean*	*SD*	*Mean*	*SD*	*Sig.*	*	*Mean*	*SD*	*Mean*	*SD*	*Sig.*	*
Transition 1	Cycling	Avg Speed Cycle	36.67	1.31	32.10	7.55	.167		34.30	5.12	30.23	2.22	0.325	
	Running	Time	85.17	16.61	96.33	13.32	.349		90.00	15.73	111.33	9.71	.010	*
		Expl. Dist.	14.33	6.16	7.40	2.25	.109		7.82	5.90	6.80	2.07	.330	
		Distance 0–15 km/h	48.47	72.55	159.70	130.75	.034	*	106.58	119.07	323.83	71.86	.331	
		Distance 15.1–20 km/h	211.72	93.21	232.30	100.01	.231		235.38	200.02	85.60	74.69	.015	*
		Distance 20.1–25 km/h	143.72	166.37	10.83	18.76	.003	*	61.44	99.06	0.00	0.00	.564	
		Distance > 25.1 km/h	1.33	3.27	0.00	0.00	.516		0.58	1.30	0.00	0.00	.439	
		HRMax	187.50	8.17	182.67	14.47	.531		183.40	15.50	188.00	23.26	.013	*
		HRAvg	182.50	7.87	179.33	14.22	.671		178.80	13.79	185.00	23.26	.024	*
		% HRMax	91.25	3.93	89.67	7.11	.671		89.40	6.90	92.50	11.63	.028	*
		Speed avg	18.07	2.18	15.25	2.10	.107		16.41	2.49	13.39	0.97	.196	
		PlayerLoad	5.32	0.55	5.80	0.59	.271		5.94	0.60	6.69	1.14	.008	*
		nStep	239.83	33.91	276.67	30.66	.159		275.20	33.80	324.67	21.08	.008	*
		tStep	332.41	13.96	336.54	19.64	.722		316.74	26.07	292.69	59.29	.883	
		tContact	224.05	30.83	254.61	14.21	.155		238.77	41.95	213.70	88.21	.883	
		tFly	108.36	29.54	81.93	33.17	.032	*	77.97	22.99	49.69	23.75	.883	
	Fatigue	CMJ	35.04	4.13	31.01	1.91	.161		28.66	2.36	24.64	0.89	.799	
Transition 2	Cycling	Avg speed cycle	35.92	1.48	34.34	1.97	.215		34.22	4.89	31.23	2.24	.959	
	Running	Time	91.17	19.70	97.67	19.66	.655		98.40	19.81	106.67	1.53	.333	
		Expl. Dist.	12.35	4.77	7.40	2.51	.143		7.86	1.60	4.10	1.87	.084	
		Distance 0–15 km/h	49.78	71.85	129.03	129.38	.023	*	138.38	150.58	285.67	82.20	.957	
		Distance 15.1–20 km/h	210.62	73.07	256.03	56.06	.025	*	229.16	42.22	119.70	5.10	.224	
		Distance 20.1–25 km/h	145.82	7.65	31.60	10.44	.031	*	52.18	5.41	0.00	0.00	.001	*
		Distance > 25.1 km/h	0.00	0.00	0.00	0.00			0.00	0.00	0.00	0.00		
		HRMax	189.17	8.82	184.33	11.55	.503		183.80	13.03	190.67	21.50	.009	*
		HRAvg	184.33	7.99	181.33	12.42	.566		179.60	11.41	187.33	22.01	.010	*
		% HRMax	92.17	4.00	90.67	6.21	.668		89.80	5.71	93.67	11.00	.069	
		Speed avg	17.87	2.00	15.60	2.24	.166		15.61	2.57	13.62	0.57	.016	*
		PlayerLoad	5.40	0.57	6.07	0.95	.221		6.00	0.71	6.69	1.06	.013	*
		nStep	240.83	32.23	283.33	57.74	.188		291.60	43.07	320.33	4.04	.016	*
		tStep	322.95	11.06	328.41	6.62	.465		336.85	11.89	325.44	26.08	.031	*
		tContact	227.03	46.79	243.55	37.42	.614		243.70	26.47	234.57	21.12	.031	*
		tFly	95.92	36.81	84.86	42.57	.679		93.15	25.49	90.86	34.17	.959	
	Fatigue	CMJ	33.15	5.46	22.45	5.63	.029	*	27.44	3.45	22.66	3.53	.001	*
Transition 3	Cycling	Avg speed cycle	35.32	1.79	34.35	2.12	.491		34.14	5.30	29.70	2.82	.325	
	Running	Time	87.33	16.97	100.33	15.82	.360		95.00	15.70	101.67	4.04	.785	
		Expl. Dist.	13.98	6.90	8.97	1.01	.265		8.20	3.53	2.57	2.66	.079	
		Distance 0–15 km/h	53.57	77.08	177.20	123.35	.001	*	146.38	170.12	298.37	68.69	.080	
		Distance 15.1–20 km/h	200.70	62.46	226.43	32.96	.171		230.14	47.10	69.43	47.05	.168	
		Distance 20.1–25 km/h	115.10	8.34	7.77	1.44	.013	*	30.10	2.37	0.00	0.00	.247	
		Distance > 25.1 km/h	0.67	1.25	0.00	0.00	.400		0.00	0.00	0.00	0.00	.006	*
		HRMax	188.50	8.04	181.00	9.64	.254		183.60	13.79	189.00	19.52	.010	*
		HRAvg	184.17	7.68	178.33	8.96	.340		179.40	12.86	185.67	20.01	.013	*
		% HRMax	92.08	3.84	89.17	4.48	.340		89.70	6.43	92.83	10.00	.018	*
		Speed avg	17.46	1.89	14.91	1.74	.092		15.70	2.44	13.02	0.69	.009	*
		PlayerLoad	5.42	0.56	6.12	1.12	.235		5.80	0.51	6.12	1.12	.009	*
		nStep	238.00	30.25	289.00	42.32	.072		287.80	36.21	290.67	37.82	.009	*
		tStep	327.65	10.44	330.21	5.47	.708		338.39	11.57	322.20	28.00	.583	
		tContact	241.74	39.45	245.28	40.56	.903		238.86	27.40	232.26	23.80	.325	
		tFly	85.90	32.37	84.93	43.99	.971		99.53	26.16	89.94	32.26	.333	
	Fatigue	CMJ	32.18	6.23	28.46	2.67	.367		29.62	2.77	23.03	6.84	.583	
Transition 4	Cycling	Avg speed cycle	35.07	2.57	34.50	2.08	.753		33.70	4.04	29.27	3.37	.060	
	Running	Time	79.83	7.83	96.00	17.52	.086		92.60	16.52	109.00	1.73	.333	
		Expl. Dist.	13.58	5.40	13.77	3.48	.960		8.58	1.88	7.63	4.33	.121	
		Distance 0–15 km/h	40.53	50.76	125.13	89.03	.004	*	121.12	166.16	258.80	103.67	.734	
		Distance 15.1–20 km/h	190.25	71.10	252.63	48.38	.334		208.24	37.02	163.80	0.63	.451	
		Distance 20.1–25 km/h	169.12	15.14	39.73	2.70	.015	*	79.98	0.58	0.00	0.00	.004	*
		Distance > 25.1 km/h	0.87	1.48	0.00	0.00	.361		0.00	0.00	0.00	0.00	.004	*
		HRMax	182.00	14.48	182.00	8.54	1000		186.20	14.52	190.67	19.01	.042	*
		HRAvg	175.33	17.03	177.33	9.02	.857		181.00	12.67	186.33	19.50	.043	*
		% HRMax	87.67	8.51	88.67	4.51	.516		90.50	6.33	93.17	9.75	.180	
		Speed avg	18.21	1.79	15.86	2.07	.119		16.25	2.52	14.05	0.53	.179	
		PlayerLoad	5.42	0.48	6.17	0.95	.143		5.84	0.51	7.04	1.14	.141	
		nStep	233.33	18.67	279.67	46.31	.061		282.00	39.96	327.67	6.81	.180	
		tStep	311.10	26.91	327.95	7.86	.337		333.00	13.94	316.99	35.08	.009	*
		tContact	223.88	46.38	241.12	37.03	.596		234.54	30.28	229.65	22.27	.009	*
		tFly	87.22	23.50	86.82	42.98	.986		98.46	33.39	87.34	37.35	.049	*
	Fatigue	CMJ	31.94	5.61	29.07	0.22	.420		27.69	4.00	22.65	4.04	.230	

* *p* < .05; Expl. Dist: explosive distance; HRMax: maximum heart rate; HRAvg: average heart rate; % HRMax: percentage of maximum heart rate; Speed Avg: average of velocity running phase; nStep: number of steps; tStep: time of step; tContact: time of contact; tFly: time of flight.

**Table 3 ijerph-19-09366-t003:** Descriptive and inferential results of the variables analyzed based on the transition in circuit with drafting.

			Elite						Amateur					
			Men		Women				Men		Women			
			*Mean*	*SD*	*Mean*	*SD*	*Sig.*	*	*Mean*	*SD*	*Mean*	*SD*	*Sig.*	*
Transition 1	Cycling	Avg speed cycle	33.94	4.04	34.10	0.42	.170		30.82	3.61	28.93	2.22	.294	
	Running	Time	73.40	5.27	103.50	19.09	.005	*	102.60	12.76	106.00	5.00	.930	
		Expl. Dist.	14.24	6.71	2.30	0.00	.712		7.88	4.90	7.03	3.12	.002	*
		Distance 0–15 km/h	11.32	2.70	229.80	264.60	.655		157.52	135.02	251.40	54.10	.002	*
		Distance 15.1–20 km/h	135.58	64.17	183.90	8.77	.041	*	268.06	16.40	167.07	13.12	.027	*
		Distance 20.1–25 km/h	250.12	200.34	0.00	0.00	.036	*	2.02	2.52	0.77	0.86	.624	
		Distance > 25.1 km/h	21.06	34.49	0.00	0.00	.136		0.00	0.00	0.00	0.00	.632	
		HRMax	189.60	9.71	185.50	10.61	.001	*	184.20	14.25	190.33	20.55	.432	
		HRAvg	184.60	10.64	181.00	9.90	.019	*	180.40	14.15	185.33	23.35	.732	
		% HRMax	92.30	5.32	90.50	4.95	.002	*	90.20	7.08	92.67	11.68	.728	
		Speed avg	20.51	1.85	14.61	2.72	.171		15.08	1.57	14.29	0.30	.262	
		PlayerLoad	5.37	0.57	6.09	1.49	.001	*	6.24	0.71	6.76	1.11	.170	
		nStep	227.60	7.92	306.50	72.83	.001	*	300.60	33.67	307.00	31.48	.167	
		tStep	328.99	4.52	317.64	38.06	.831		315.43	28.42	319.29	7.66	.738	
		tContact	228.05	23.20	229.09	79.44	.807		245.28	20.62	264.47	8.33	.737	
		tFly	100.93	19.18	88.55	41.37	.831		70.15	23.48	54.82	0.66	.738	
	Fatigue	CMJ	34.27	6.66	20.07	1.05	.029	*	29.24	5.33	22.21	5.75	.688	
Transition 2	Cycling	Avg speed cycle	33.42	3.38	33.45	0.35	.032	*	30.44	2.70	29.07	0.64	.201	
	Running	Time	78.40	5.90	110.50	26.16	.340		70.80	56.57	98.33	7.09	.334	
		Expl. Dist.	14.52	11.23	6.35	5.02	.342		10.36	7.76	4.33	4.23	.943	
		Distance 0–15 km/h	17.60	6.53	234.75	275.98	.019	*	183.58	120.42	243.67	28.05	.598	
		Distance 15.1–20 km/h	127.68	41.86	199.25	69.65	.874		234.26	19.01	144.93	6.45	.396	
		Distance 20.1–25 km/h	270.66	29.90	5.65	3.99	.008	*	18.38	5.70	0.00	0.00	.677	
		Distance > 25.1 km/h	10.32	15.46	0.00	0.00	.005	*	0.16	0.36	0.00	0.00	.679	
		HRMax	191.60	8.73	190.00	12.73	.216		186.20	15.06	191.67	19.60	.636	
		HRAvg	186.80	7.98	184.50	12.02	.276		179.20	17.20	185.67	21.20	.634	
		% HRMax	93.40	3.99	92.25	6.01	.129		89.60	8.60	92.83	10.60	.508	
		Speed avg	19.77	0.94	14.58	3.58	.001	*	14.43	1.58	14.06	0.35	.239	
		PlayerLoad	5.36	0.60	6.52	1.45	.001	*	6.39	0.55	6.38	1.54	.296	
		nStep	235.60	13.22	323.00	69.30	.001	*	312.20	34.05	281.33	22.37	.256	
		tStep	329.50	7.03	325.98	12.76	.058		311.69	30.22	323.92	1.12	.164	
		tContact	232.94	20.69	228.32	66.64	.065		230.39	33.74	272.90	6.26	.159	
		tFly	96.56	15.38	97.66	53.88	.081		81.30	32.85	51.01	5.50	.179	
	Fatigue	CMJ	34.50	3.24	23.97	3.60	.032	*	29.60	4.60	28.08	6.33	.668	
Transition 3	Cycling	Avg speed cycle	33.68	3.74	32.65	2.47	.478		30.68	3.66	28.93	1.66	.294	
	Running	Time	75.60	5.98	112.00	24.04	.179		106.20	17.98	97.67	2.08	.131	
		Expl. Dist.	13.62	4.04	8.60	3.68	.014	*	4.66	2.27	6.67	3.33	.583	
		Distance 0–15 km/h	12.96	4.77	244.50	250.74	.003	*	178.88	101.34	185.33	12.82	.599	
		Distance 15.1–20 km/h	119.42	44.31	190.00	43.77	.894		241.66	39.97	221.00	23.65	.464	
		Distance 20.1–25 km/h	10.68	17.24	0.00	0.00	.091		0.00	0.00	0.00	0.00	.422	
		Distance > 25.1 km/h	193.20	9.01	190.50	12.02	.002	*	186.20	13.59	192.67	19.60	.460	
		HRMax	186.80	9.73	185.50	12.02	.068		177.60	15.84	185.33	20.40	.393	
		HRAvg	93.40	4.87	92.75	6.01	.016	*	88.80	7.92	92.67	10.20	.404	
		% HRMax	20.16	1.36	14.17	3.31	.005	*	14.39	1.42	14.81	0.02	.410	
		Speed avg	5.35	0.39	6.35	1.35	.003	*	6.19	0.95	6.63	1.29	.708	
		PlayerLoad	230.40	16.98	326.50	68.59	.006	*	305.60	45.22	291.00	28.58	.709	
		nStep	330.52	7.44	322.34	2.74	.686		317.10	16.31	306.93	17.68	.863	
		tStep	217.04	30.81	200.89	18.49	.478		249.46	24.38	226.95	16.90	.294	
		tContact	113.48	24.22	121.45	15.76	.329		67.64	11.64	79.98	1.54	.334	
		tFly	32.18	3.87	19.43	1.24	.686		28.82	5.62	22.27	5.05	.863	
	Fatigue	CMJ	32.98	3.47	32.70	0.57	.041	*	30.48	3.07	28.97	1.36	.887	
Transition 4	Cycling	Avg speed cycle	75.20	4.32	111.00	32.53	.329		97.40	12.64	104.67	6.66	.334	
	Running	Time	15.40	7.71	7.70	0.42	.413		8.40	6.33	9.50	3.84	.272	
		Expl. Dist.	18.84	5.82	215.65	270.75	.776		158.74	89.87	197.37	34.51	.396	
		Distance 0–15 km/h	113.88	28.00	176.80	57.84	.980		242.04	28.63	216.77	13.40	.288	
		Distance 15.1–20 km/h	35.84	22.84	41.15	16.26	.058		4.56	1.84	1.13	0.67	.778	
		Distance 20.1–25 km/h	59.12	79.53	0.00	0.00	.042	*	0.00	0.00	0.00	0.00	.778	
		Distance > 25.1 km/h	194.20	8.98	190.50	13.44	.332		186.20	11.86	192.00	18.08	.606	
		HRMax	187.80	9.34	185.00	12.73	.382		179.80	12.79	186.67	20.21	.603	
		HRAvg	93.90	4.67	92.50	6.36	.128		89.90	6.40	93.33	10.10	.740	
		% HRMax	20.14	1.65	14.86	3.88	.001	*	14.90	1.01	14.40	0.46	.251	
		Speed avg	5.74	0.35	6.34	1.53	.001	*	6.01	1.04	6.95	1.15	.246	
		PlayerLoad	236.00	12.35	321.00	82.02	.001	*	287.80	44.58	302.67	41.43	.246	
		nStep	327.13	4.17	332.78	11.24	.174		307.44	27.67	312.25	6.03	.018	*
		tStep	224.13	22.59	233.35	69.28	.133		234.73	32.03	254.31	0.85	.090	
		tContact	101.00	17.13	99.44	58.03	.070		72.71	34.39	57.94	6.87	.899	
		tFly	32.98	3.47	32.70	0.57	.041	*	30.48	3.07	28.97	1.36	.887	
	Fatigue	CMJ	30.74	6.05	21.22	0.86	.285		24.68	5.97	20.25	5.18	.376	

* *p* < .05; Expl. Dist: explosive distance; HRMax: maximum heart rate; HRAvg: average heart rate; % HRMax: percentage of maximum heart rate; Speed Avg: average of velocity running phase; nStep: number of steps; tStep: time of step; tContact: time of contact; tFly: time of flight.

**Table 4 ijerph-19-09366-t004:** Descriptive and inferential results depending on the competitive level and use, or absence, of drafting.

			Elite						Amateur					
			without Drafting		with Drafting				without Drafting		with Drafting			
			Mean	SD	Mean	SD	Sig.	*	Mean	SD	Mean	SD	Sig.	*
Transition 1	Cycling	Avg speed cycle	35.14	4.53	33.99	3.31	.055		28.44	3.27	27.98	2.69	.033	
	Running	Time	88.89	15.74	82.00	17.18	.417		93.18	16.52	93.67	17.48	.936	
		Expl. Dist.	12.02	6.09	10.83	8.00	.738		9.86	5.78	9.09	6.21	.716	
		Distance 0–15 km/h	85.54	10.32	73.74	15.18	.855		133.78	13.35	137.20	143.02	.945	
		Distance 15.1–20 km/h	218.58	22.23	149.39	19.63	.000	*	192.48	5.53	187.08	9.01	.984	
		Distance 20.1–25 km/h	99.42	15.02	178.66	20.66	.000	*	61.11	8.61	45.03	7.22	.057	
		Distance > 25.1 km/h	0.89	2.67	15.04	3.00	.025	*	1.65	5.88	14.90	37.97	.262	
		HRMax	185.89	9.99	188.43	9.25	.611		185.53	13.43	187.40	12.69	.689	
		HRAvg	181.44	9.58	183.57	9.74	.669		181.29	12.87	182.87	13.40	.737	
		% HRMax	90.72	4.79	91.79	4.87	.669		90.65	6.43	91.43	6.70	.737	
		Speed avg	17.13	2.46	18.82	3.44	.788		21.41	2.77	21.48	3.25	.692	
		PlayerLoad	5.48	0.57	5.58	0.84	.655		271.71	41.63	278.33	47.41	.676	
		nStep	252.11	35.96	250.14	49.07	.972		16.26	2.59	16.67	3.20	.697	
		tStep	333.79	14.92	325.74	16.91	.883		321.52	31.12	321.01	19.61	.981	
		tContact	234.24	29.64	228.35	37.56	.580		231.94	43.95	241.22	30.67	.982	
		tFly	99.55	31.54	97.39	23.81	.558		89.58	23.95	79.80	30.67	.033	*
	Fatigue	CMJ	33.69	3.95	30.21	8.82	.329		39.84	3.83	47.65	6.35	.508	
Transition 2	Cycling	Avg speed cycle	35.39	1.72	33.43	2.76	.222		29.23	3.12	28.51	3.01	.145	
	Running	Time	93.33	18.70	87.57	19.56	.702		94.03	4.57	91.92	3.69	.777	
		Expl. Dist.	10.70	4.68	12.19	10.20	.129		41.19	17.59	42.39	25.94	.239	
		Distance 0–15 km/h	76.20	9.48	79.64	15.48	.072		87.00	42.98	100.07	8.59	.256	
		Distance 15.1–20 km/h	225.76	20.08	148.13	14.95	.023	*	37.33	3.42	153.99	64.73	.041	*
		Distance 20.1–25 km/h	107.74	14.97	194.94	19.67	.032	*	263.68	57.42	105.35	14.89	.039	*
		Distance > 25.1 km/h	0.00	0.00	7.37	13.59	.726		128.82	46.40	44.10	10.65	.164	
		HRMax	187.56	9.37	191.14	8.86	.396		187.00	12.25	189.60	12.63	.142	
		HRAvg	183.33	8.99	186.14	8.23	.902		182.94	11.86	183.73	13.74	.583	
		% HRMax	91.67	4.49	93.07	4.12	.605		96.21	1.19	91.87	6.87	.599	
		Speed avg	17.11	2.25	18.29	3.02	.950		20.97	2.19	21.65	3.53	.218	
		PlayerLoad	5.62	0.74	5.69	0.96	.412		5.77	0.46	5.28	0.46	.412	
		nStep	255.00	43.98	260.57	52.30	.058		260.55	24.51	261.55	29.89	.416	
		tStep	324.77	9.74	328.49	7.94	.964		328.44	14.11	321.98	18.65	.912	
		tContact	232.54	42.27	231.62	32.10	.812		236.18	33.91	239.46	32.75	.330	
		tFly	92.23	36.48	96.88	25.34	.077	*	92.26	19.80	82.51	32.70	.355	
	Fatigue	CMJ	29.58	7.42	31.49	5.97	.557		24.72	5.32	29.32	4.43	.404	
Transition 3	Cycling	Avg speed cycle	34.99	1.83	33.39	3.26	.487		33.81	3.70	31.59	3.49	.121	
	Running	Time	91.67	16.87	86.00	20.87	.785		92.43	3.24	92.56	2.99	.183	
		Expl. Dist.	12.31	6.02	12.19	4.37	.633		40.01	24.55	42.05	23.57	.410	
		Distance 0–15 km/h	94.78	10.65	79.11	15.25	.886		93.82	60.10	85.73	4.90	.709	
		Distance 15.1–20 km/h	209.28	20.01	139.59	15.41	.032	*	187.89	18.64	267.47	2.52	.021	*
		Distance 20.1–25 km/h	302.02	14.82	198.71	22.46	.028	*	91.37	14.65	60.13	10.41	.049	*
		Distance > 25.1 km/h	0.44	1.04	7.63	15.01	.351		130.59	50.11	156.80	40.59	.863	
		HRMax	186.00	8.82	192.43	8.94	.480		185.82	11.73	190.40	12.30	.247	
		HRAvg	182.22	8.09	186.43	9.36	.207		182.00	11.35	183.27	13.64	.272	
		% HRMax	91.11	4.04	93.21	4.68	.079		94.65	1.13	92.27	1.36	.396	
		Speed avg	16.61	2.15	18.45	3.41	.244		20.61	3.13	21.26	3.10	.917	
		PlayerLoad	5.65	0.79	5.64	0.80	.451		5.71	40.57	280.40	50.66	.778	
		nStep	255.00	40.87	257.86	56.35	.523		275.71	23.81	266.37	3.10	.919	
		tStep	328.50	8.78	328.18	7.36	.295		330.30	14.30	320.24	14.73	.706	
		tContact	242.92	37.24	212.43	27.42	.010	*	239.84	31.12	227.68	28.61	.710	
		tFly	85.58	33.74	115.75	21.15	.027	*	90.45	21.12	92.56	28.61	.110	
	Fatigue	CMJ	30.94	5.43	28.54	7.00	.884		28.34	4.88	28.47	5.63	.740	
Transition 4	Cycling	Avg speed cycle	34.88	2.30	32.90	2.85	.390		32.16	3.23	31.83	3.16	.205	
	Running	Time	85.22	13.43	85.43	22.23	.441		91.59	15.52	93.27	18.12	.922	
		Expl. Dist.	13.64	4.61	13.20	7.33	.357		11.09	46.62	10.86	6.47	.251	
		Distance 0–15 km/h	68.73	7.34	75.07	14.65	.737		117.68	12.66	107.42	12.02	.246	
		Distance 15.1–20 km/h	10.67	10.29	21.08	5.29	.029	*	201.88	21.44	202.65	88.49	.765	
		Distance 20.1–25 km/h	311.87	35.10	280.21	14.82	.041	*	54.39	10.94	85.85	13.44	.031	*
		Distance > 25.1 km/h	0.58	1.25	42.23	7.11	.396		0.31	0.23	19.71	5.14	.018	
		HRMax	182.00	12.22	193.14	9.34	.391		184.76	13.56	190.60	11.61	.023	
		HRAvg	176.00	14.23	187.00	9.33	.617		179.29	14.33	184.53	12.43	.682	
		% HRMax	88.00	7.12	93.50	4.66	.521		89.65	7.16	92.27	6.21	.678	
		Speed avg	17.43	2.11	18.63	3.31	.318		21.32	2.92	22.23	3.23	.884	
		PlayerLoad	5.67	0.71	5.91	0.75	.170		5.96	0.87	5.15	0.95	.255	
		nStep	248.78	35.93	260.29	54.25	.402		264.85	23.40	256.54	3.02	.265	
		tStep	316.72	23.21	328.74	6.35	.161		321.55	23.02	318.34	18.59	.394	
		tContact	229.63	41.97	226.76	34.06	.380		231.08	34.31	234.93	30.07	.157	
		tFly	87.09	28.41	100.55	27.52	.043	*	90.48	11.29	83.42	11.48	.194	
	Fatigue	CMJ	30.98	4.66	28.02	6.79	.963		28.02	4.49	25.13	5.86	.466	

* *p* < .05; Expl. Dist: explosive distance; HRMax: maximum heart rate; HRAvg: average heart rate; % HRMax: percentage of maximum heart rate; Speed Avg: average of velocity running phase; nStep: number of steps; tStep: time of step; tContact: time of contact; tFly: time of flight.

## Data Availability

MDPI Research Data Policies.

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
