# Peer review of "Does the Physiological Response of a Triathlete Change in the Use or Absence of Drafting?"

_ijerph, 2022, doi:10.3390/ijerph19159366_

Round 1

Reviewer 1 Report

Thank you for the opportunity to review “Does the physiological response of a triathlete change in the use/absence of drafting/without drafting?”. I read this manuscript with interest. Here, the authors study differences between sexes in their physiological response to multiple transitions of bike-run similar to triathlon training and competition was analyzed.

I have several concerns about this manuscript, which are described below.

Consider revising the title. The multiple “/” make it difficult to understand. Also, this study does not seem to only focus on drafting, but the title makes it seem that way.

Introduction:

Consider using sex throughout the manuscript to describe males and females. Gender (and using women/men) does not seem to play a role in this study, and these are distinct identities. https://orwh.od.nih.gov/sex-gender#:~:text=%22Sex%22%20refers%20to%20biological%20differences,across%20societies%20and%20over%20time.

As I see it, this study has three main focuses: 1. Competition level, 2. Sex, and 3. Drafting/not drafting. Each of these should be clearly described and justified in the introduction. For example, consider describing drafting in more detail- what is this? How common is it? What is known about the physiological effects of this- what research has already been done? Differences between sexes is also a major part of this manuscript but isn’t described in the introduction. Consider revising to include this.

The purpose of the study do not seem to match the results well. Consider revising. How are you characterizing the physical demands?

Methods

Can you justify the adult group, or why there is a combination of adults and minors in the amateur group?

2.5 Why were t-tests performed if the data was not normally distributed? This appears to be hundreds of t-tests based on your tables. Can you justify this? Do you have the number of participants to support this multitude of comparisons?

2.5.1- Was test 2 done 12 weeks later?

Results

Table 1- Does the amateur group include both age groups?

            What is ‘tContacto’ and ‘tVuelo’ or ‘playerload’?

            It isn’t clear enough what is compared here. Males and females just in elite? Or male elite and male amateur?

Table 3- Is this collapsed across sexes? This should be clearer. It seems that you’ve shown some differences between sexes in your other tables, so I don’t understand why you’d collapse them here.

Discussion

This opening paragraph seems like it could be in the introduction instead of discussion. Perhaps summarizing the study here would be more helpful.

4.1 line 260- I’m not sure how anatomical differences are related to your results here. There are many factors beyond the Q- angle (hormones, body composition, training level etc), especially since this study is more about physiology than kinematics.

Line 269- I found this paragraph hard to understand.

Line 305- This is a 5 line sentence. It was hard to follow. Consider revising.

Many references in your discussion should have the author’s name rather than citation number. For example, line 316: ‘To do this, Gutierrez et al. analyzed the aerodynamic loss…[32]’

I found this discussion overall to be hard to understand and not clearly describing or interpreting the results presented.

Conculsion

Line 346- ‘Drafting has a greater influence on the biomechanics of women than in men…’ I don’t understand where this comes from.

Author Response

All the reviewer's suggestions have been addressed.
All actions taken are presented in the attached document.

Reviewer 2 Report

This manuscript offers interesting information for triathlon coaches. I provide some suggestions:

The introduction could be improved. A more detailed description/explanation of drafting, advantages and disadvantages is welcomed.

Similarly, a description of the characteristics and importance of multi-transition training could be of help when it comes to justify the aim of the research.

How were the participants recruited? All of the participants who were contacted finished the study? Please, clarify.

I found the participants characteristics difficult to follow. My advise is to include a table indicating age/category/level/sex.

Any chance to provide “the n” on all the tables?

A short paragraph indicating whether significant differences between categories (i.e. U16 v U18) existed would be welcomed.

What are the best advices that can be given to coaches based on the results of research. The authors describe some interesting ideas (lines 341-349), while other ones (lines 350-355) add nothing to the current body of knowledge.

Limitations should be placed at the end of the discussion section.

The covid rule (line 363). Is still compulsory? If not, there is no need to mention it.

Author Response

(The authors gave the same response as above.)

Reviewer 3 Report

Dear authors,

Thank you for this paper, I consider it could be very interesting to the scientific community.

Some concerns about this article:

Abstract: include the age of the participants (they seem adults), as well as data on the results (tests performed, significance...),

Introduction: the theoretical framework is very simple, with only 6 references, this must be improved with aspects on the technique of cycling, other sports where drafting can influence (such as running) or aspects related to studies who have analyzed these aspects in cycling or different modalities of drafting  (Hausswirth et al. 2001)

2.2. Sample: specify "the position of your category" (line 82) the fourth... the sixth...? in what competitions? Why you choose young triathlete and not adults?

Were aspects such as food, rest... or protocols prior to conducting the study taken into account? (to ensure that sessions are held under the same conditions)

The study sample is quite small, I suggest including it in limitations

The conclusions, title and abstract should be changed taking into account the age of the sample (at least, conclusions and title)

Bibliographical references must be translated into English

Author Response

(The authors gave the same response as above.)

Reviewer 4 Report

The references in the introduction can be improved if the authors consider more actual papers around the subject.

The Statistical analysis must be described and improved with other authors with more references that only Field (2009).

The results must be clarified. For example, if the “Avg Speed Cycle” is 15 km/h what is the best answer Distance 0-15 km/h or Distance 15-20km/h? The same question when the “Avg Speed Cycle” is 20 km/h.

Please, improve the conclusion considering the results obtained. 

Author Response

(The authors gave the same response as above.)

Round 2

Reviewer 2 Report

I have read the authors' responses. I have no further suggestions to make.

Reviewer 3 Report

I consider the authors have made a high effort to improve the manuscript and it is adequate to be published

Reviewer 4 Report

Congratulations on the result obtained in the last version.